# Nurse-led secondary preventive follow-up after stroke/TIA and ACS for patients aged 80 years or older: A post-hoc analysis of the randomized controlled NAILED trial

Karl Ingard, Anna-Lotta Irewall, Thomas Mooe, Joachim Ögren*

Department of Public Health and Clinical Medicine, Umeå University, Östersund, Sweden

* joachim.ogren@umu.se

## Abstract

### Background

The evidence supporting secondary prevention with antihypertensives and lipid-lowering drugs after cerebrovascular disease or acute coronary syndrome (ACS) is not as strong for persons aged ≥80 years. The Nurse-based, Age-independent Intervention to Limit Evolution of Disease (NAILED) trial was a randomized controlled trial in which secondary preventive follow-up with titration of antihypertensives and lipid-lowering drugs was compared to usual care. In this substudy, we investigated the efficacy and safety of the NAILED intervention in persons aged ≥80 years.

### Methods and findings

Patients admitted to Östersund Hospital with ACS, stroke, or transient ischemic attack between 2010 and 2014 were randomized to a nurse-led telephone-based follow-up (intervention group) or usual care (control group) and followed from discharge until 31 December 2017, with a maximum follow-up of 5 years. This post-hoc analysis included all patients aged ≥80 years (n = 394). The primary outcome was a composite of cardiovascular death, myocardial infarction, and stroke. The secondary endpoints were cardiovascular death, myocardial infarction, stroke, ischemic stroke, all-cause mortality, fracture, orthostatic hypotension, serious bleeding, and health-related quality of life. During a mean follow-up of 3.8 years, 31.7% (n = 64) of the patients in the intervention group and 37.5% (n = 72) in the control group reached the primary endpoint (HR 0.82, 95% CI 0.58–1.14, P = 0.23). The risk of cardiovascular death was significantly decreased (HR 0.64, 95% CI 0.41–0.998, P = 0.049) and the risk of fracture non-significantly increased (HR 1.47, 95% CI 0.95–2.27, P = 0.08) in the intervention group compared to the control group.

**Data availability statement:** As open access to individual-level data was not specified in the original application approved by the ethics committee, sharing the data freely would violate the ethical approval. Therefore, the underlying data is only available upon reasonable request. Data are available from the Institutional Data Access at Department of Public Health and Clinical Medicine, Umeå University (contact via registrator@umu.se) for researchers who meet the criteria for access to confidential data.

**Funding:** The study received funding from the Unit of Research, Development and Education, Region Jämtland Härjedalen and the Swedish Heart–Lung Foundation. The funders had no role in the study design, data collection and analysis, decision to publish, or preparation of the manuscript.

**Competing interests:** The authors have declared that no competing interests exist.

## Conclusions

The intervention in the NAILED trial did not reduce the risk of major cardiovascular events in patients aged ≥80 years. The trends of lower risk of cardiovascular events and increased risk of fractures need to be validated in future research.

## Trial registration

ISRCTN23868518, ISRCTN96595458.

The NAILED risk factor trial is registered in the ISRCTN registry, (ISRCTN23868518) for the stroke/TIA and (ISRCTN96595458) for the ACS cohort. The strict ICMJE requirement of prospective registration of clinical trials came to our attention when the recruitment had already begun. The study was therefore retrospectively registered on 19 June 2012. We confirm that all related and ongoing trials are now registered.

## Introduction

Coronary artery disease (CAD) and cerebrovascular disease are major causes of morbidity and mortality among the elderly, especially in the growing population of persons aged ≥80 years [1]. Secondary prevention with antihypertensives and lipid-lowering drugs reduces the risk of new cardiovascular events [2–4]. However, the mean ages of participants in most randomized controlled trials (RCTs) have been lower than those of patients with CAD and cerebrovascular disease in clinical practice [5], with resulting uncertainty regarding the risk-benefit ratio of treatment among the elderly.

Based on a meta-analysis including a mix of primary and secondary preventive trials, lipid-lowering drugs reduce the risk of new cardiovascular events among patients aged ≥75 years [6]. In primary prevention, one RCT showed that antihypertensives are also effective among persons older than 80 years [7]. However, the individuals in these RCTs were relatively healthy, limiting the generalizability of their results to the frail elderly population with comorbidities [8,9]. The prevalence of frailty, comorbidity, and polypharmacy increases with age, and they constitute risk factors for adverse effects from medications [10]. RCTs on secondary prevention of cardiovascular disease with antihypertensives are scarce [11]. Lipid-lowering drugs, such as statins, are likely to be safe in the elderly [12,13]. Adverse effects of antihypertensives have been found in elderly patients with hypotensive symptoms [14], but conflicting results have been reported regarding an increased risk [15] or no increased risk [14] of falls. Observational studies have suggested that frailty may diminish the mortality-reducing effect of antihypertensives for individuals aged ≥65 years [16] and increase the risk of mortality for those aged ≥70 years [17].

The European Society of Cardiology´s 2021 guidelines on secondary cardiovascular prevention among the elderly share many similarities with the recommendations for younger individuals. According to these guidelines, the treatment goals for low-density lipoprotein cholesterol (LDL-C) are a serum level <1.4 mmol/L and ≥50%

reduction compared to baseline regardless of age. For blood pressure (BP), the treatment goals for persons aged ≥70 years is to achieve systolic blood pressure (SBP) <140 mmHg and diastolic blood pressure (DBP) <80 mmHg, and then to reduce the SBP to 130 mmHg if tolerated. However, the guidelines include a reservation that the evidence for the strict BP targets is less strong among those aged ≥80 years, especially patients who are frail [18].

In the Nurse-based, Age-independent Intervention to Limit Evolution of Disease (NAILED) trial, a systematic nurse and telephone-based follow-up after acute coronary syndrome (ACS), stroke, or transient ischemic attack (TIA) yielded improved BP and LDL-C levels [19,20] and a decreased risk of new cardiovascular events compared to usual care. The NAILED trial was a pragmatic study that aimed to include a relatively unselected population commonly seen in clinical practice [21]. In this sub-study, we aimed to assess the effect of the intervention in the NAILED trial on recurrent cardio-vascular events in participants ≥80 years of age and to explore whether the intervention increased the risk of adverse effects in this subgroup.

## Materials and methods

### Study design

This study was a non-prespecified post-hoc analysis of the randomized controlled NAILED trial, focusing on participants aged 80 years or older.

### Participants

All patients hospitalized at Östersund Hospital for stroke or TIA between 1 January 2010 and 31 December 2013 and ACS between 1 January 2010 and 31 December 2014 were screened for inclusion. Stroke included ischemic and hemorrhagic events but not subarachnoid hemorrhage. ACS included myocardial infarction (MI) and unstable angina. Only participants aged ≥80 years at discharge from the initial hospitalization were included in this post-hoc analysis of the NAILED trial. Patients who were unable to communicate with a telephone, for example, due to hearing disability, cognitive impairment, or aphasia, were excluded. In addition, patients with severely impaired health who were not considered for secondary pre-ventive treatment and patients already participating in an incompatible clinical trial were excluded from this analysis [21].

### Setting

Östersund Hospital is the only hospital in the county of Jämtland-Härjedalen. Approximately 126,500 inhabitants lived in the county at the beginning of the study; 35.0% of them lived in the city of Östersund and the remainder resided in villages or rural areas. All patients with suspected stroke, TIA, or MI, except a few in terminal care, were referred to Östersund Hospital. Twenty-eight primary healthcare centers provided primary care [21].

### Randomization

The randomized allocation to the intervention or control group (1:1) was computer-generated. The participants, caregivers, and study team were not blinded to the allocation, but the physicians who assessed the outcome events were blinded. The randomization process was described in detail previously [21].

### Intervention group

The intervention group received a nurse-led, telephone-based follow-up that included counseling to improve modifiable risk factors and assess pharmacological treatment. Follow-up was performed 1 and 12 months after discharge from the initial hospitalization, and subsequent follow-ups were performed yearly until termination of the intervention on 31 December 2017. Lifestyle counseling included the following points: to remain physically active by exercising a minimum of 30 minutes per day, 5 days a week; smoking cessation; and encouragement to adhere to the Swedish Food Agency's

recommended diet, including an increased intake of vegetables, fruits, fiber, and unsaturated fats and a decreased intake of saturated fats. Before each follow-up call, BP and blood lipids were measured at the closest healthcare facility. If the participants did not reach the LDL-C and BP targets, a study physician was contacted to adjust the pharmacological treatment. These adjustments were individualized, and no fixed algorithm or protocol was used. In these cases, a new contact was established 4 weeks after the adjustments to assess the effect of the treatment. Further adjustments could be made, if necessary, until the participant reached the target levels or when further adjustments were considered impossible [21].

### Control group

The control group received usual care. Secondary prevention began during the initial hospitalization. Patients with ACS usually had a follow-up visit with a cardiology nurse 1 month post-discharge and with a cardiologist 2–3 months post-discharge. Patients with stroke or TIA were referred to primary care upon discharge. The general practitioner usually had primary responsibility for the long-term follow-up. The general practitioners were supplied with the BP and blood lipid measurements and, therefore, had the opportunity to act accordingly [21].

### Target levels

The target BP and LDL-C levels were based on local guidelines, and the same target BP and LDL-C levels were used independently of group allocation. The target BP was < 140/90 mmHg throughout the study period. The target LDL-C level was initially <2.5 mmol/L, but subsequent changes were made to match changes in local guidelines. On 31 March 2013, the target LDL-C level was lowered to <1.8 mmol/L for patients with diabetes, and this target level was adopted for all study participants on 1 January 2017 [21].

### Outcomes

The primary endpoint was a composite of cardiovascular death, type 1 MI, and stroke. Stroke included ischemic and hemorrhagic events but not subarachnoid hemorrhage. The secondary endpoints were any of the individual components of the primary endpoint, ischemic stroke, all-cause mortality, fractures, serious bleeding, orthostatic hypotension (OH), and health-related quality of life (HRQoL). All fractures were included (International Classification of Diseases 10th revision codes S0270, S0290, S12, S22, S32, S42, S52, S62, S72, S82, S92, T02-P, and T0890). Serious bleeding was defined as intracranial hemorrhage, bleeding that required hospital admission, or bleeding that required transfusion or surgery. OH was defined as a ≥ 20 mmHg decrease in SBP or ≥10 mmHg decrease in DBP 1 minute after standing from a seated position. A definition of cardiovascular death is available in S1 File Outcome definition. HRQoL was measured using the EuroQoL 5 dimensions 3 levels (EQ-5D-3L) questionnaire including the EQ-5D descriptive system and a visual analog scale (EQ-VAS). The EQ-5D-3L descriptive system covers five dimensions of health (mobility, self-care, usual activities, pain/discomfort, and anxiety/depression), and each dimension is graded on a three-level scale (1 = no problem, 2 = some problems, 3 = extreme problems), generating a number sequence called a health profile. Each profile was then converted to a health index between 0 and 1 (0 = dead, 1 = perfect health) using weights provided by the Swedish experience-based value set in Burström et al. [22]. With the EQ-VAS, health was rated by the participants on a vertical visual analog scale on which 0 represented the worst health and 100 the best imaginable health. Cardiovascular prognosis and mortality were prespecified outcomes [23,24], and the collection of EQ-5D-3L was a part of the prespecified health economic analysis.

### Data collection

Patients were followed for 5 years after discharge, until 31 December 2017 or death, whichever occurred first. Baseline characteristics were collected in-hospital by study nurses through interviews and reviews of the medical records. BP and blood lipids were measured and an orthostatic test performed prior to each follow-up [19,20]. The EQ-5D-3L was

distributed as a postal survey prior to each follow-up during the first 2 years of follow-up and then, afterwards, conducted as part of the telephone follow-up. Study physicians who were unaware of the participants' study group assignment, assessed all potential outcome events based on predefined outcome criteria. To identify events, all of the participants´ discharge records at the Department of Internal Medicine at Östersund Hospital during the follow-up were scrutinized. In addition, a local patient register was searched for defined diagnosis codes to identify outcome events at other hospital departments or, for fractures, primary care. All events identified through diagnosis codes were confirmed through reviews of the medical records. Patients who moved out of the county were considered lost to follow-up at the time they moved because their medical records became inaccessible [21].

## Statistical analysis

All statistical analyses were conducted according to the intention to treat principle. Independent samples T-tests, the Mann-Whitney U-test, and chi-squared were used to compare baseline characteristics, mean BP and LDL-C values, and HRQoL outcomes between the groups. The group median EQ-5D-3L and mean EQ-VAS values were based on the mean of all available values during follow-up for each participant. The alpha level was set to 0.05. Kaplan-Meier survival analysis was performed to present the cumulative incidence for short-term (365 days) and long-term (1825 days) follow-up, with a log-rank test for comparisons between the randomized groups. Day 0 was set as the day of discharge from the initial hospitalization. Reasons for censoring in the survival analyses were migration out of the county, death during follow-up (if not included in the outcome), and followed until the end of follow-up (i.e., 365 days in the short-term analysis and 1825 days, or the number of days between discharge and 31 December 2017, in the long-term analysis). Hazard ratios were calculated for the outcomes using univariable Cox proportional hazard regressions. Log(−log) plots were used to evaluate the proportional hazard assumption in the Cox models. We did not correct for multiplicity. In an elderly population, dropout from active study participation during long-term follow-up may be considerable due to health deterioration associated with aging. For the primary outcome events and fractures, follow-up was also performed among those who discontinued active participation, thereby avoiding the risk of attrition bias. Collection of the secondary outcome variables (OH and EQ5D) did, however, require active participation. We did not use imputation to replace values missing due to discontinuation, as it was reasonable to assume that discontinuation was often due to deterioration in health and that the available data were therefore not necessarily predictive of the missing data. A comparison of baseline characteristics and primary outcome events among those who discontinued active participation preterm (<3 years) to those who participated for at least 3 years is available in S1 Table. An effort was made to investigate competing risks. However, the construction of the composite variable made that impossible, and the numbers were too small to show any effect for the individual variables. IBM SPSS Statistics version 29 was used for the statistical analyses.

## Trial registration

### Ethics

All eligible patients were informed of the study, and those who chose to participate provided written consent. The NAILED trial was approved by the Regional Ethics Committee, Umeå University (reference numbers Dnr 09-142M and Dnr 13–204-32M) [21]. The study protocol approved by the ethics committee is available in S2 File Study protocol.

## Results

### Participants

A total of 1167 patients, comprising 36.2% of all patients assessed for participation in the NAILED trial, were aged ≥80 years at hospital discharge. Among these, 33.8% (n = 394) were included and randomized into the intervention group (n = 202) and control group (n = 192). The most common reasons for non-participation were unwillingness to participate

(n = 211, 18.1%), death during the acute phase (n = 137, 11.7%), and cognitive impairment (n = 134, 11.5%; Fig 1). Baseline characteristics are provided in Table 1. Most characteristics were well-balanced between the groups. The median age was 84.0 years and 48.7% of the participants were women. Those who were excluded had a higher median age, included a higher proportion of women, and had a higher incidence of stroke as the qualifying event than those who were included (S3 Table).

**BP and LDL-C**

The mean values and the number and proportion of patients reaching target levels of SBP, DBP, and LDL-C during follow-up are presented in Table 2. A numerical decrease in mean LDL-C and SBP values occurred in the intervention group during follow-up, but the corresponding measures remained at the same level or increased in the control group. Thus, SBP and LDL-C were numerically lower in the intervention group than the control group at each yearly follow-up

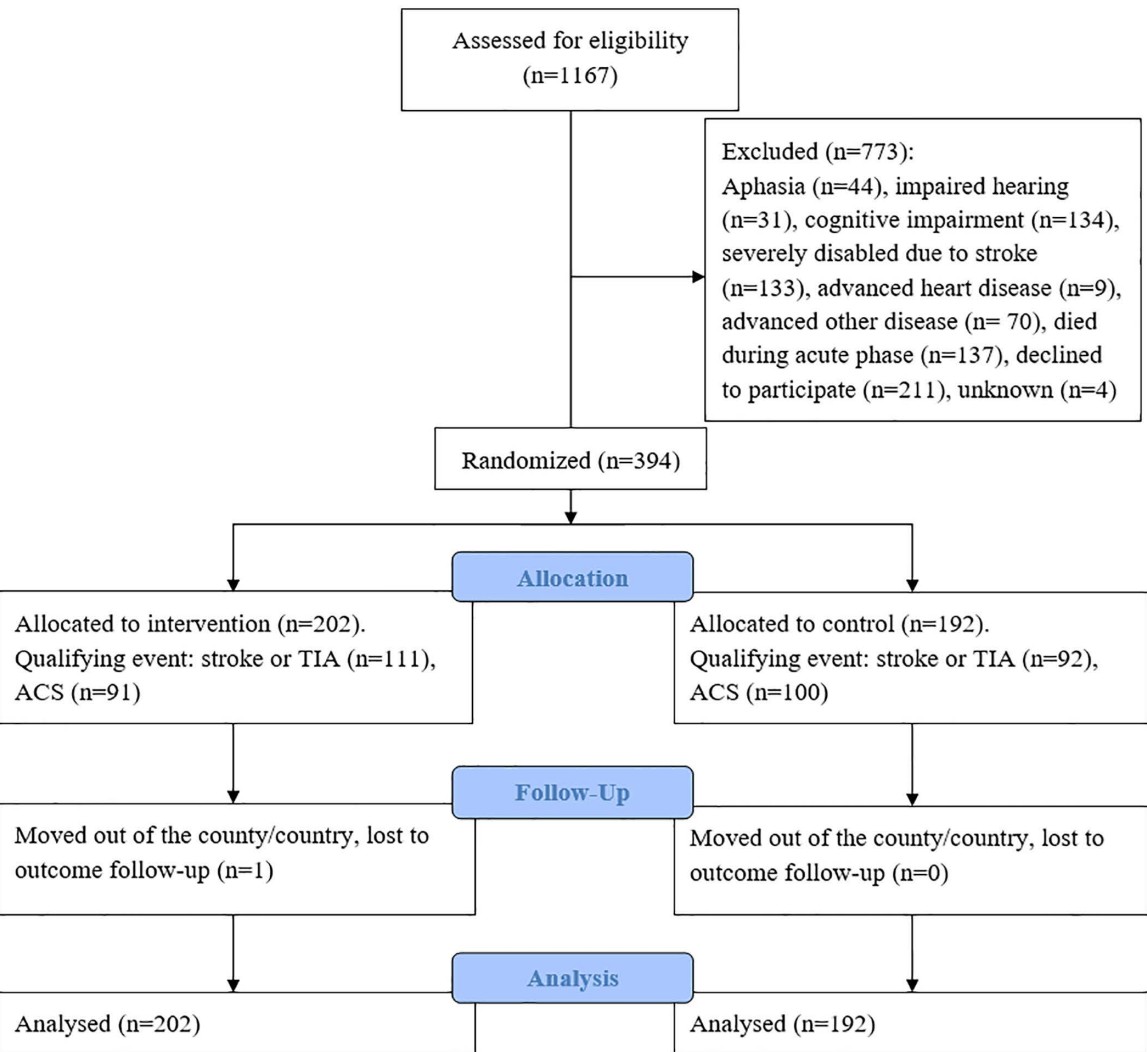

**Fig 1. Study flow chart.** TIA, transient ischemic attack; ACS, acute coronary syndrome.

**Table 1. Baseline Characteristics of Included Patients.**

| | Intervention | Control |
|---|---|---|
| N (%) | 202 (51.3) | 192 (48.7) |
| Women | 101 (50.0) | 91 (47.4) |
| Age, years | 84.0 (81.0-86.0) | 84.0 (81.0-88.0) |
| Age<85, years | 117 (57.9) | 110 (57.3) |
| Low education level[a] | 138 (68.3) | 151 (78.6) |
| Current/former smoker[b] | 88 (43.8) | 77 (40.1) |
| **Qualifying event** | | |
| Unstable angina | 4 (2.0) | 6 (3.1) |
| Myocardial infarction | 87 (43.1) | 94 (49.0) |
| Ischemic stroke | 68 (33.7) | 50 (26.0) |
| Intracerebral hemorrhage | 6 (3.0) | 5 (2.6) |
| TIA | 37 (18.3) | 37 (19.3) |
| **Baseline measurements** | | |
| BMI, kg/m$^2$ | 25.2 (23.0-27.7) | 24.8 (22.4-27.9) |
| eGFR, ml/min | 53.7 (44.0-66.1) | 51.9 (41.0-64.3) |
| LDL-C, mmol/L[c] | 2.9 (2.2-3.6) | 2.8 (2.2-3.5) |
| **Medical history** | | |
| Atrial fibrillation | 64 (31.7) | 48 (25.0) |
| Ischemic heart disease[d] | 72 (35.6) | 75 (39.1) |
| Peripheral artery disease | 9 (4.5) | 7 (3.6) |
| Diabetes | 39 (19.3) | 42 (21.9) |
| Congestive heart failure | 16 (7.9) | 20 (10.4) |
| COPD | 5 (2.5) | 15 (7.8) |
| Hypertension | 149 (73.8) | 126 (65.6) |
| Stroke | 33 (16.3) | 23 (12.0) |
| TIA | 17 (8.4) | 7 (3.6) |
| **Medications at discharge** | | |
| Antihypertensive drug | 178 (88.1) | 175 (91.1) |
| −1 drug | 42 (20.8) | 54 (28.1) |
| −2 drugs | 81 (40.1) | 66 (34.4) |
| -≥3 drugs | 55 (27.2) | 55 (26.8) |
| Lipid-lowering drug | 132 (65.3) | 136 (70.8) |
| Antiplatelet drug | 157 (77.7) | 164 (85.4) |
| Warfarin | 44 (21.8) | 27 (14.1) |

Categorical values are presented as N (%) and quantitative values as median (interquartile range). Significant differences between the groups: low education level ($P=0.02$), COPD ($P=0.02$), previous TIA ($P=0.048$), antiplatelet drug ($P=0.049$), and warfarin ($P=0.046$). TIA, transient ischemic attack; BMI, body mass index; eGFR, estimated glomerular filtration rate; LDL-C, low-density lipoprotein cholesterol; COPD, chronic obstructive pulmonary disease.

[a]Low education level refers to <10 years of education. [b]1 missing value in the intervention group. [c]6 missing values in the control group and 9 missing values in the intervention group. [d]Ischemic heart disease was defined as a history of myocardial infarction, angina, coronary artery bypass grafting, or percutaneous coronary intervention.

**Table 2. Blood Pressure and LDL-C During Follow-up.**

| | Intervention | | | Control | | |
|---|---|---|---|---|---|---|
| | Mean (SD) | Target level, N (%) | Nª | Mean (SD) | Target level, N (%) | Nª |
| **Systolic blood pressure, mmHg** | | | | | | |
| 1 month | 137.9 (18.3) | 101 (54.0) | 187 | 137.5 (19.4) | 83 (47.7) | 174 |
| 12 months | 134.0 (17.7) | 101 (61.6) | 164 | 140.0 (19.2) | 69 (47.3) | 146 |
| 24 months | 134.5 (17.5) | 78 (59.1) | 132 | 140.5 (20.8) | 56 (44.4) | 126 |
| 36 months | 133.6 (16.9) | 74 (65.5) | 113 | 138.6 (20.0) | 60 (51.7) | 116 |
| 48 months | 128.0 (15.6) | 68 (74.7) | 91 | 139.1 (20.9) | 40 (48.2) | 83 |
| 60 months | 133.5 (17.7) | 40 (64.5) | 62 | 137.0 (21.0) | 26 (54.2) | 48 |
| **Diastolic blood pressure, mmHg** | | | | | | |
| 1 month | 74.7 (11.2) | 161 (86.1) | 187 | 75.7 (11.2) | 147 (84.5) | 174 |
| 12 months | 73.3 (10.6) | 151 (92.1) | 164 | 76.7 (12.0) | 121 (82.9) | 146 |
| 24 months | 73.5 (11.2) | 119 (90.2) | 132 | 78.1 (12.2) | 102 (81.0) | 126 |
| 36 months | 72.9 (10.2) | 106 (93.8) | 113 | 76.7 (11.2) | 99 (85.3) | 116 |
| 48 months | 70.8 (9.5) | 87 (95.6) | 91 | 77.1 (11.5) | 69 (83.1) | 83 |
| 60 months | 73.4 (10.5) | 58 (93.5) | 62 | 75.8 (13.6) | 38 (79.2) | 48 |
| **LDL-C, mmol/L** | | | | | | |
| 1 month | 2.52 (0.8) | 95 (51.1) | 186 | 2.32 (0.9) | 100 (57.8) | 173 |
| 12 months | 2.25 (0.9) | 110 (69.2) | 159 | 2.44 (1.1) | 84 (58.7) | 143 |
| 24 months | 2.16 (1.0) | 100 (76.9) | 130 | 2.42 (1.1) | 77 (60.2) | 128 |
| 36 months | 2.26 (1.0) | 70 (63.6) | 110 | 2.38 (1.1) | 70 (60.9) | 115 |
| 48 months | 2.20 (0.9) | 59 (61.5) | 96 | 2.43 (1.1) | 42 (46.2) | 91 |
| 60 months | 2.31 (1.1) | 36 (48.6) | 74 | 2.47 (1.1) | 28 (41.8) | 67 |

Mean values and the number and proportion of patients reaching target levels during follow-up. LDL-C, low-density lipoprotein cholesterol; SD, standard deviation.

ªNumber of patients with a valid measurement.

throughout the study period. The proportion of participants reaching target SBP, DBP, and LDL-C levels was higher in the intervention group than in the control group in all follow-ups after the initial follow-up at 1 month.

## Primary and secondary endpoints

During a mean follow-up of 3.8 years, 31.7% (n = 64) of the participants in the intervention group and 37.5% (n = 72) of the participants in the control group reached the primary composite endpoint. The results of the outcome analyses for the primary and secondary endpoints are presented in Fig 2, Table 3, and S2 Table. The difference in the primary composite endpoint between the intervention and control groups was not significant. Regarding the secondary outcomes, the incidence of cardiovascular death was significantly lower in the intervention group, whereas all-cause mortality did not differ significantly between the groups (Table 3 and S2 Table). There was a trend towards an increased incidence of fractures in the intervention group compared to the control group (Table 3), a difference that became significant when the follow-up period was limited to 1 year (S2 Table). Nine of the 51 fractures in the intervention group and 4 of the 34 fractures in the control group occurred within 60 days after a follow-up. The incidence of OH was significantly lower in the intervention group than in the control group, during both short-term (S2 Table) and long-term follow-up (Table 3). We found no significant difference in the incidence of serious bleeding (Table 3 and S2 Table). The median EQ-5D-3L index values were lower in the intervention group than in the control group (0.84 [IQR 0.76–0.90] vs. 0.86 [IQR 0.77–0.92], P = 0.046), whereas the mean EQ-VAS scores did not differ between the groups (63.91 [SD 14.7] vs. 64.25 [SD 14.4], P = 0.82).

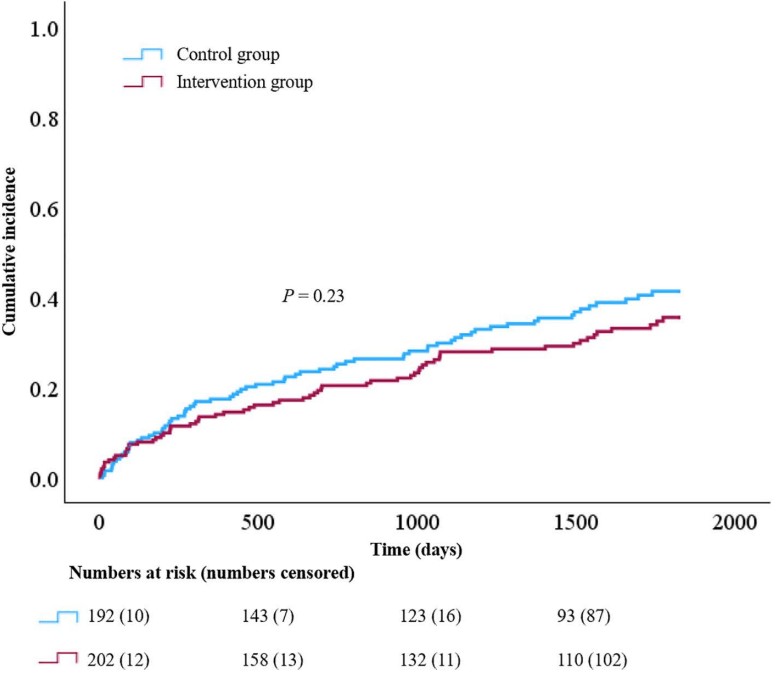

**Fig 2. Kaplan-Meier curve of the primary outcome.** Cumulative incidence of cardiovascular death, myocardial infarction, and stroke (primary endpoint) during 5 years of follow-up.

**Table 3. Number of Patients Who Reached the Primary and Secondary Endpoints During Follow-up.**

| | Intervention, N (%) | Control, N (%) | Absolute difference (%) | HR (95% CI) | P value |
|---|---|---|---|---|---|
| **Primary endpoint** | | | | | |
| CV death, MI, or stroke | 64 (31.7) | 72 (37.5) | − 5.8 | 0.82 (0.58-1.14) | 0.23 |
| -CV death[a] | 16 (25.0) | 21 (29.2) | | | |
| -MI[a] | 19 (29.7) | 22 (30.6) | | | |
| -Stroke[a] | 29 (45.3) | 29 (40.3) | | | |
| **Secondary endpoints** | | | | | |
| CV death | 32 (15.8) | 46 (24.0) | − 8.2 | 0.64 (0.41-0.998) | 0.049 |
| MI | 19 (9.4) | 22 (11.5) | − 2.1 | 0.79 (0.43-1.46) | 0.45 |
| Stroke | 29 (14.4) | 33 (17.2) | − 2.8 | 0.80 (0.49-1.32) | 0.39 |
| All-cause mortality | 77 (38.1) | 82 (42.7) | − 4.6 | 0.86 (0.63-1.17) | 0.33 |
| Ischemic stroke | 27 (13.4) | 27 (14.1) | − 0.7 | 0.92 (0.54-1.56) | 0.75 |
| Fracture | 51 (25.2) | 34 (17.7) | 7.5 | 1.47 (0.95-2.27) | 0.08 |
| Orthostatic hypotension | 73 (36.1) | 85 (44.3) | − 8.2 | 0.70 (0.51-0.95) | 0.02 |
| Serious bleeding | 19 (9.4) | 19 (9.9) | − 0.5 | 0.91 (0.48-1.72) | 0.77 |

HR, hazard ratio; CV death, cardiovascular death; MI, myocardial infarction.

[a]The components of the primary endpoint are presented as proportions (%) of the primary endpoint.

As shown in Fig 3, the between-group difference in median EQ-5D-3L index values resulted from a decrease in median index values in the intervention group after 2 years, which was driven mainly by a difference in self-care (S3 Table). The distribution of EQ-5D levels for each dimension at 1 month, 3 years, and 5 years is presented in S3 Table. The proportion

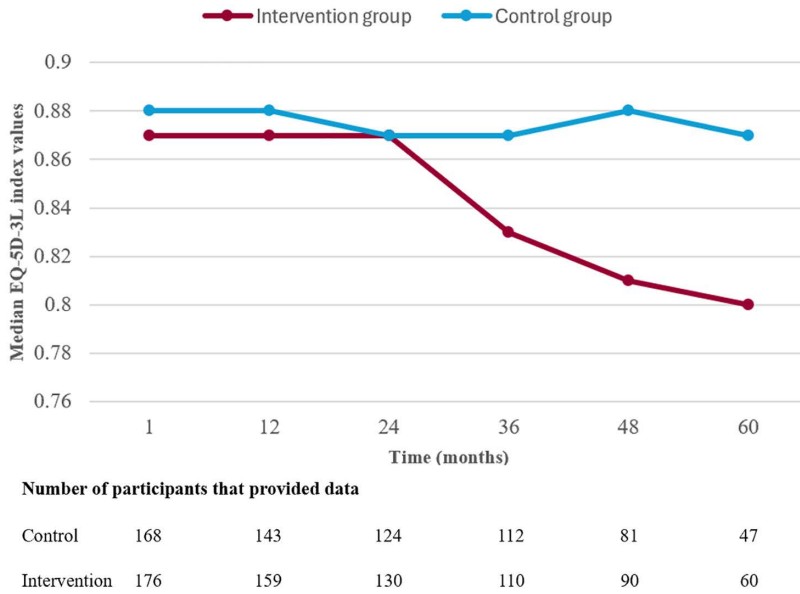

**Fig 3. Median EQ-5D-3L index values during 5 years of follow-up.**

of participants that provided EQ-5D-3L data beyond 2 years was similar between the groups, but a large proportion of participants with a non-fatal primary outcome component event or a fracture did not provide EQ-5D-3L data after these events (primary outcome: 53/99, 53.5%; fracture: 45/85, 52.9%), with a relatively larger proportion of missing data within the control group than the intervention group (S4 Table).

### Subgroup analyses

A new stroke event was a common component of the primary outcome among those with stroke or TIA as the qualifying event, whereas cardiovascular death and MI were more common among those with ACS as the qualifying event. Subgroup analyses of the primary endpoint and its components are presented in Table 4 and the remaining secondary endpoints in Table 5. The numerical, but non-significant, difference in the primary endpoint in favor of the intervention group was numerically more pronounced in the stroke/TIA subgroup, among women, and among those ≤85 years old.

## Discussion

In this post-hoc analysis of the NAILED trial, we investigated the efficacy and safety of a nurse-led secondary preventive follow-up after ACS, stroke, or TIA in patients aged ≥80 years compared to usual care. During 5 years of follow-up, fewer patients in the intervention group reached the primary composite endpoint, but this trend was not significant due primarily to a decreased risk of cardiovascular death. However, we also observed a trend towards an increased risk of fractures in the intervention group, mainly driven by women who were qualified due to stroke or TIA. Several of the differences between the intervention and control groups were present already after 1 year of follow-up.

Similar interventions have not shown compelling results of a reduced risk of recurrent cardiovascular events in the elderly. In the Drugs and Evidence-Based Medicine in the Elderly (DEBATE) study, patients in Helsinki aged ≥75 years with previous cardiovascular disease were randomized to either a geriatrician-led nonpharmacological and pharmacological follow-up or usual care. During a mean follow-up of 3.4 years, the intervention had no beneficial effect on recurrent cardiovascular events or mortality. Importantly, the DEBATE study included home-dwelling patients with previous CAD,

**Table 4. Subgroup Analysis of the Occurrence of the Primary Endpoint and Its Components.**

| | Primary endpoint | Primary endpoint components | | |
|---|---|---|---|---|
| | | CV death | MI | Stroke |
| **Intervention group** | | | | |
| ACS | 29 (31.9) | 17 (18.7) | 13 (14.3) | 8 (8.8) |
| Stroke or TIA | 35 (31.5) | 15 (13.5) | 6 (5.4) | 21 (18.9) |
| Women | 33 (32.7) | 16 (15.8) | 9 (8.9) | 15 (14.9) |
| Men | 31 (30.7) | 16 (15.8) | 10 (9.9) | 14 (13.9) |
| Age < 85 years | 29 (24.8) | 13 (11.1) | 7 (6.0) | 15 (12.8) |
| Age ≥ 85 years | 35 (41.2) | 19 (22.4) | 12 (14.1) | 14 (16.5) |
| **Control group** | | | | |
| ACS | 35 (35.0) | 29 (29.0) | 14 (14.0) | 9 (9.0) |
| Stroke or TIA | 37 (40.2) | 17 (18.5) | 8 (8.7) | 24 (26.1) |
| Women | 36 (39.6) | 18 (19.8) | 10 (11.0) | 19 (20.9) |
| Men | 36 (35.6) | 28 (27.7) | 12 (11.9) | 14 (13.9) |
| Age < 85 years | 37 (33.6) | 20 (18.2) | 7 (6.4) | 22 (20.0) |
| Age ≥ 85 years | 35 (42.7) | 26 (31.7) | 15 (18.3) | 11 (13.4) |

Data are presented as N (%). CV, cardiovascular; MI, myocardial infarction; ACS, acute coronary syndrome; TIA, transient ischemic attack.

cerebrovascular disease, or peripheral artery disease regardless of when it had occurred [25]. In contrast, the NAILED study recruited patients during their hospitalization for the index event. The incidence of recurrent cardiovascular events and mortality is particularly high in the first 6 months after an index event [26]. Therefore, the cardiovascular risk profile could have differed between the two studies. In addition, the treatment goals for LDL-C were less ambitious, as the DEBATE study had a goal of <3 mmol/L [25]. The Coronary Infarction Follow-up in the Elderly (KORINNA) study was an RCT that enrolled patients aged ≥65 years who were hospitalized due to MI. A nurse-based follow-up with home visits and telephone calls that included BP control and medication adherence was compared to usual care. During a follow-up

**Table 5. Subgroup Analysis of the Occurrence of Secondary Endpoints.**

| | All-cause mortality | Ischemic stroke | Fracture | Orthostatic hypotension | Serious bleeding |
|---|---|---|---|---|---|
| **Intervention group** | | | | | |
| ACS | 37 (40.7) | 7 (7.7) | 20 (22.0) | 40 (44.0) | 6 (6.6) |
| Stroke or TIA | 40 (36.0) | 20 (18.0) | 31 (27.9) | 33 (29.7) | 13 (11.7) |
| Women | 35 (34.7) | 13 (12.9) | 33 (32.7) | 33 (32.7) | 9 (8.9) |
| Men | 42 (41.6) | 14 (13.9) | 18 (17.8) | 40 (39.6) | 10 (9.9) |
| Age < 85 years | 36 (30.8) | 14 (12.0) | 27 (23.1) | 45 (38.5) | 13 (11.1) |
| Age ≥ 85 years | 41 (48.2) | 13 (15.3) | 24 (28.2) | 28 (32.9) | 6 (7.1) |
| **Control group** | | | | | |
| ACS | 49 (49.0) | 7 (7.0) | 18 (18.0) | 49 (49.0) | 10 (10.0) |
| Stroke or TIA | 33 (35.9) | 20 (21.7) | 16 (17.4) | 36 (39.1) | 9 (9.8) |
| Women | 35 (38.5) | 14 (15.4) | 19 (20.9) | 35 (38.5) | 9 (9.9) |
| Men | 47 (46.5) | 13 (12.9) | 15 (14.9) | 50 (49.5) | 10 (9.9) |
| Age < 85 years | 30 (27.3) | 18 (16.4) | 15 (13.6) | 48 (43.6) | 10 (9.1) |
| Age ≥ 85 years | 52 (63.4) | 9 (11.0) | 19 (23.2) | 37 (45.1) | 9 (11.0) |

Data are presented as N (%). ACS, acute coronary syndrome; TIA, transient ischemic attack.

of 3 years, the intervention did not significantly reduce the time to unplanned readmission or death. The KORINNA study differed substantially from this study in the chosen outcomes and participant age. In addition, the KORINNA study was restricted to patients with MI [27]. Thus, an appropriate comparison to our study could be with the group with ACS as the index event, which had a comparably higher hazard ratio for the primary outcome compared to the stroke/TIA group in the subgroup analysis.

In summary, the NAILED, DEBATE, and KORINNA interventions all found reduced BP and LDL-C in their respective populations of elderly patients, which was significant in the case of the DEBATE trial. However, none of the interventions could translate into a conclusive and clinically relevant effect on cardiovascular events [25,27]. Such effects could have been expected, as the association between an improved cardiovascular risk profile and reduced risk of cardiovascular disease in secondary prevention among persons ≥75 years of age has been found previously in the case of lipid-lowering drugs [6], and less evidently for antihypertensives [11,14]. In the total NAILED trial, the intervention achieved a significant reduction in BP and LDL-C compared to usual care [19,20], despite the control group performing better than in the EUROASPIRE IV survey in the ACS cohort [20]. In the current trial, the lack of significance may be attributed to the underpowered nature of this post-hoc analysis. We observed a consistent numerical trend in favor of the intervention for all cardiovascular outcomes and the hazard ratio for the primary outcome was very similar to the hazard ratio in the total NAILED [21].

In contrast to previous observational studies in frail elderly populations, we found no evidence of a diminished mortality-reducing effect of antihypertensives [16] or increased risk of mortality [17].

During follow-up, the mean SBP, DBP, and LDL-C were approximately 5−6 mmHg, 3−4 mmHg, and 0.1–0.2 mmol/L, respectively, lower in the intervention group than in the control group. The intervention group also had a higher proportion of participants achieving target levels, and the follow-up time was longer than in similar trials [25,27]. The BP reduction in the current trial can be compared to a meta-analysis of RCTs, where a reduction of SBP by 5 mmHg in patients with previous cardiovascular disease, with a mean age of 65 years, reduced the risk of major cardiovascular events by about 10% [2]. The clinical relevance of the small numerical difference in LDL-C in the current trial is uncertain, but a reduction of LDL-C of 1 mmol/L has previously been shown to reduce the risk of major cardiovascular events in older adults by 26% [6]. The mechanism we propose for the favorable trend in cardiovascular outcome is the improved cardiovascular risk factor profile in the intervention group, which was achieved through the structured and repeated titration of the pharmacological treatment to reach the target BP and LDL-C values. Having an MI or stroke after 80 years of age is often associated with a short life expectancy [28,29]. Therefore, a shorter time perspective may sometimes be more relevant in clinical decision-making. Interestingly, the trend of the main results in this study was evident after 1 year and, although these results are even more underpowered, they suggest that secondary prevention with antihypertensives and lipid-lowering drugs may also be beneficial for the elderly in the short-term.

The NAILED trial started recruitment in 2010, and the patients were followed until 2017. Therefore, the intervention and the control group participants likely did not receive the aspects of secondary cardiovascular prevention introduced in the guidelines after 2017. These include long-term secondary prevention with dual antiplatelet therapy, the lipid-lowering drugs proprotein convertase subtilisin/kexin type 9 inhibitors and inclisiran, and adding a glucagon-like peptide-1 receptor agonist or sodium-glucose cotransporter 2 inhibitor for patients with diabetes mellitus with established atherosclerotic cardiovascular disease [18].

During 5 years of follow-up, we observed a difference in the HRQoL with regard to the EQ-5D-3L index values, favoring the control group, whereas the EQ-VAS scores did not differ significantly between the allocation groups. This proposed difference in HRQoL is small, and it is uncertain to what extent it could be attributed to the intervention, as we also saw differences in baseline characteristics, distribution of outcome events, and possibly selective dropout, which could have biased the results. An RCT that compared intensive antihypertensive treatment (SBP goal of <120 mmHg) vs. standard treatment (SBP goal of <140 mmHg) in participants ≥80 years of age in primary prevention assessed HRQoL using the

Veterans RAND 12-Item Health Survey. In the intensively treated group, no difference was found in the Mental Component Summary and a small decline was observed in the Physical Component Summary [30]. Furthermore, an observational study found no association between SBP values and EQ-5D-3L index values in patients aged ≥75 years who were treated with antihypertensives [31]. In our study, more participants in the intervention group than the control group reported experiencing problems with mobility, usual activities, and self-care. The difference in perceived self-care increased from the 3-year follow-up. The proportion of participants that continued to provide EQ-5D-3L data after a primary outcome and fracture event was lower in the control group than in the intervention group. This means that the between-group difference in outcome events was not reflected well in the EQ-5D-3L data, and selective drop-out probably constituted a bias in the result.

The trend of increased risk of fractures in the intervention group in this study warrants caution, but the mechanism is not entirely evident. Differences in baseline characteristics, including an increased frequency of women and previous stroke in the intervention group compared to the control group may have contributed because both characteristics constitute risk factors for fractures [32] and we did not adjust for these differences in the analysis. Previous studies on the association between antihypertensive treatment and fractures have had conflicting results and OH, which is associated with falling [33], was rarer in the intervention group than the control group. Observational studies suggest that the use of antihypertensives among the elderly contributes to falls [34] and fractures [15], whereas other observational studies [35,36] and an RCT [14] found no such association. Kahlaee et al. [37] conducted a meta-analysis of mainly observational data from persons ≥60 years of age. They found a significantly elevated risk of falling within the first 24 hours after initiating, changing, or increasing the dose of antihypertensives. This increased risk remained significant for diuretics until day 21. Chronic use of antihypertensives, independent of type, was not associated with a significantly increased risk of falling. The mechanism the authors proposed for this acutely increased risk of falling was a drop in BP and OH possibly caused by hypovolemia, bradycardia, or arterial or venous dilatation. In the NAILED study, no fixed algorithm or protocol was used for pharmacological treatment and, thus, both diuretics, which can cause hypovolemia, and beta-blockers, which can cause bradycardia, were allowed as part of the antihypertensive treatment based on individual assessment and prompt initiation or dose escalation in response to a BP above the target level. The difference in the incidence of fractures observed in our study seems, at least in part, to have arisen early during the follow-up as the difference reached significance when the follow-up was limited to 1 year, but we saw no clear pattern of an acutely elevated risk of falling following the escalation of antihypertensive treatment compared to the control group. Data on the occurrence of OH was collected yearly, and a transient episode could have been missed. The increased incidence of OH during the follow-up and decreased incidence of fractures in the control group compared to the intervention group suggests that asymptomatic OH alone may not signify that an adjustment of pharmacological treatment is necessary.

The strengths of this study include the fact that the age-based exclusion criterion was determined at a late age. To the best of our knowledge, no directly comparable studies have had a higher age for participants. Although 773 of 1167 patients were excluded from the analysis, the study population was relatively unselected because the exclusion criteria were kept at a minimum. The patients included in this study likely represented a population of elderly patients for whom strict secondary preventive treatment is often considered but not always initiated, often due to a fear of adverse effects. Reasonably, these factors increased the external validity. The single-center design constituted a strength in that Östersund Hospital was the only referral center for patients with suspected stroke, TIA, and ACS, which facilitated the sensitivity in the collection of outcome events. Another strength was the relatively long follow-up. However, the present study also had several limitations. First, the single-center design may have reduced the external validity. Second, occasional outcome events could have been missed if the patients were solely treated in primary care or at other hospitals. Third, the general practitioners were supplied with the BP and LDL-C measurements and could act accordingly. This likely represented more frequent measurements than normally conducted in usual care, possibly leading to underestimating the results. Fourth, despite patients with aphasia and hearing disability

possibly benefitting from the intervention, they were excluded because of their inability to use a telephone. It is possible that customized means of communication could have rendered these patients able to participate. Fifth, acute kidney injury and acute renal failure are potential complications in the elderly patients intensively treated with anti-hypertensives [30], but this was not analyzed in the current trial because data on renal function was not collected. Sixth, one possible explanation for the lack of significance in the outcomes may have been a lack of power in the data because of too few study participants. Finally, post-hoc analyses are associated with an inherently increased rate of type 1 errors due to multiplicity [38], which we did not correct for because it could have made the results unfavorably conservative given the exploratory nature of this post-hoc analysis.

## Conclusions

In conclusion, the use of a nurse-led telephone-based secondary preventive follow-up including counseling to improve modifiable risk factors and titration of antihypertensives and lipid-lowering drugs to target levels did not reduce the risk of a composite of cardiovascular death, MI, and stroke for persons aged ≥80 years, but there was a trend in favor of the intervention and, in the case of cardiovascular death, the difference reached significance. Although there is a need for sufficiently powered replication studies to confirm these exploratory results, the trend of a lower incidence of cardiovascular diseases suggests that this kind of intervention may also be beneficial for elderly patients, except for a possibly increased risk of fractures.

## Supporting information

**S1 File. Outcome definition. Definition of cardiovascular death.**
(DOCX)

**S1 Table. Comparison of patients who discontinued active participation before 3 years to those who participated for at least 3 years.** Categorical values are presented as N (%) and quantitative values as median (interquartile range). TIA, transient ischemic attack.
(DOCX)

**S2 File. Study protocol.**
(DOCX)

**S2 Table. Baseline characteristics of the excluded group.** Categorical values are presented as N (%) and quantitative values as median (interquartile range). TIA, transient ischemic attack; BMI, body mass index; eGFR, estimated glomerular filtration rate; LDL-C, low-density lipoprotein cholesterol; COPD, chronic obstructive pulmonary disease. [a]Low education level refers to <10 years of education. [b]Ischemic heart disease was defined as a history of myocardial infarction, angina, coronary artery bypass grafting, or percutaneous coronary intervention.
(DOCX)

**S3 Table. Number of patients who reached the primary and secondary endpoints with a maximal follow-up of 1 year.** HR, hazard ratio; CV, cardiovascular; MI, myocardial infarction.
(DOCX)

**S4 Table. Number of EQ-5D-3L dimensions and levels during follow-up.** Values are presented as N (%).
(DOCX)

**S5 Table. Number of participants who provided EQ-5D-3L data.** Values are given as N (%). [*]Any time point before the event. Some events occurred before the first follow-up (1 month). [**]Any time point after the event.
(DOCX)

**S1 Checklist. CONSORT 2010 checklist of information to include when reporting a randomised trial\*.**
(DOCX)

## Acknowledgments

We would like to thank the study personnel who conducted the NAILED trial and provided us with the data and Lars Söderström for helping with the statistical analyses.

## Author contributions

**Formal analysis:** Karl Ingard, Joachim Ögren.

**Funding acquisition:** Thomas Mooe.

**Methodology:** Anna-Lotta Irewall, Thomas Mooe, Joachim Ögren.

**Writing – original draft:** Karl Ingard.

**Writing – review & editing:** Anna-Lotta Irewall, Thomas Mooe, Joachim Ögren.

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
