## [Decision Letter · Decision Letter 0]

5 Jun 2025

Dear Dr. Ögren,

Thank you for submitting your manuscript to PLOS ONE. After careful consideration, we feel that it has merit but does not fully meet PLOS ONE’s publication criteria as it currently stands. Therefore, we invite you to submit a revised version of the manuscript that addresses the points raised during the review process.

We look forward to receiving your revised manuscript.

Kind regards,

Rizaldy Taslim Pinzon

Academic Editor

PLOS ONE

Journal Requirements:

2. We note that you have selected “Clinical Trial” as your article type. PLOS ONE requires that all clinical trials are registered in an appropriate registry (the WHO list of approved registries is at      https://www.who.int/clinical-trials-registry-platform/network/primary-registries""
https://www.who.int/clinical-trials-registry-platform/network/primary-registries and more information on trial registration is at http://www.icmje.org/about-icmje/faqs/clinical-trials-registration/ ).

Please state the name of the registry and the registration number (e.g. ISRCTN or ClinicalTrials.gov ) in the submission data and on the title page of your manuscript.

a) Please provide the complete date range for participant recruitment and follow-up in the methods section of your manuscript.

b) If you have not yet registered your trial in an appropriate registry, we now require you to do so and will need confirmation of the trial registry number before we can pass your paper to the next stage of review. Please include in the Methods section of your paper your reasons for not registering this study before enrolment of participants started. Please confirm that all related trials are registered by stating: “The authors confirm that all ongoing and related trials for this drug/intervention are registered”.

Please see http://journals.plos.org/plosone/s/submission-guidelines#loc-clinical-trials for our policies on clinical trials.

3. In the online submission form, you indicated that [As open access to individual-level data was not specified in the original application approved by the ethics committee, the underlying data is only available upon reasonable request. Data are available from the Institutional Data Access at Department of Public Health and Clinical Medicine, Umeå University (contact via registrator@umu.se) for researchers who meet the criteria for access to confidential data.].

Additional Editor Comments:

Thank you authors for the submission.

I proceed the comments from the reviewers.

Reviewers' comments:

Reviewer's Responses to Questions

**Comments to the Author**

1. Is the manuscript technically sound, and do the data support the conclusions?

Reviewer #1: Yes

Reviewer #2: Yes

2. Has the statistical analysis been performed appropriately and rigorously?

Reviewer #1: Yes

Reviewer #2: Yes

3. Have the authors made all data underlying the findings in their manuscript fully available?

Reviewer #1: Yes

Reviewer #2: Yes

4. Is the manuscript presented in an intelligible fashion and written in standard English?

Reviewer #1: Yes

Reviewer #2: Yes

Reviewer #1: As the statistical reviewer I will focus on methods and reporting. There is balance in the two arms in terms of baseline characteristics, as expected, so univariable approaches are fine.

Major

1) The authors say "No method was used to impute missing values". Why is that? Power is modest as it is (see next major comment) so why not try to at least paritally address that using multiple imputation? A clear explanation is needed.

2) The results need to be interpreted more cautiously since the analyses are unerpowered to detect anything but very large effects. This is already discussed as limitation in the relevant section.

Minor

1) was the proportional hazards assumption evaluated in the Cox models?

2) more common to say "univariable" rather than "univariate" analyses

3) the issue with the competing risks approach was not clear. I don't see why the competing risk of all-cause death could not be computed and if findings are similar, so be it (and for the best), as the authors imply.

Reviewer #2: An interesting sub-analysis of the NAILED trial of secondary prevention after ACS and stroke looking at the effects of intensive secondary prevention in the over 80 age group.

1. Well described and presented.

2. The secondary prevention measures are limited by 2025 standards and dont include aspects such as prolonged dual anti-platelet therapy, low dose anti-coagulants, SGLTI and GLP1 analogues in diabetics and weight management. Were PCSK9I or inclisiran used in either group? Comments in discussion please.

3. The effects of intervention will be dependent not only on the intervention efficacy, but on the standard treatment. Can the authors provide information on how these patients are managed in 'standard care'. Are targets for LDL and BP used in Sweden and how much intervention is typically provided by 'standard' primary care? In an 'excellent' primary care system, improvements may be hard to achieve.

4. To me, the effects of the intervention were very modest with a BP difference for sys and dia of only about 3-4 mmHg and an LDL difference of about 0.2 mmol/l. Can the authors provide data on how many patients achieved LDL targets of 1.8 (or 1.4) in each group? Would such small differences over a fairly short follow up period be expected to reduce events? For such small differences the study was probably under powered. Comment please in discussion.

5. Do we have any indirect measures of the efficacy of the lifestyle measures? Weight change in each group would be an important metric.

6. The data on fractures are intriguing and well discussed. I agree that differences at baseline may have caused this apparent effect.

**Do you want your identity to be public for this peer review?** For information about this choice, including consent withdrawal, please see our Privacy Policy

Reviewer #1: No

Reviewer #2: **Yes: ** Jonathan N Townend

---

## [Author Response · Author response to Decision Letter 1]

2 Jul 2025

COMMENT: Please ensure that your manuscript meets PLOS ONE's style requirements, including those for file naming.

RESPONSE: The file naming of the supporting information files has been changed to meet PLOS ONE’s style requirements by replacing the underscore characters with spaces. Changes to the supporting information captions were also made to match the names and titles of the supporting files (pages 27 and 28).

COMMENT: Please state the name of the registry and the registration number (e.g. ISRCTN or ClinicalTrials.gov) in the submission data and on the title page of your manuscript.

RESPONSE: The ISRCTN registry number has now been added to the abstract in accordance with PLOS ONE’s policies on clinical trials (page 3, second paragraph): ISRCTN23868518, ISRCTN96595458.

COMMENT: Please provide the complete date range for participant recruitment and follow-up in the methods section of your manuscript.

RESPONSE: The complete data range for participant recruitment and follow-up is provided on page 5, third paragraph, and page 6, third paragraph.

COMMENT: If you have not yet registered your trial in an appropriate registry, we now require you to do so and will need confirmation of the trial registry number before we can pass your paper to the next stage of review. Please include in the Methods section of your paper your reasons for not registering this study before enrolment of participants started. Please confirm that all related trials are registered by stating: “The authors confirm that all ongoing and related trials for this drug/intervention are registered”.

RESPONSE: The NAILED risk factor trial is registered in the ISRCTN registry. The reason for why the trial was retrospectively registered is given on page 10, second paragraph. We added a sentence regarding trial registration of ongoing and related trials (page 11, first paragraph): “We confirm that all related and ongoing trials are now registered.”

COMMENT: In the online submission form, you indicated that [As open access to individual-level data was not specified in the original application approved by the ethics committee, the underlying data is only available upon reasonable request. Data are available from the Institutional Data Access at Department of Public Health and Clinical Medicine, Umeå University (contact via registrator@umu.se) for researchers who meet the criteria for access to confidential data.].

RESPONSE: We have now updated our statement, motivating why we cannot make our data freely available: “As open access to individual-level data was not specified in the original application approved by the ethics committee, sharing the data freely would violate the ethical approval. Therefore, the underlying data is only available upon reasonable request. Data are available from the Institutional Data Access at Department of Public Health and Clinical Medicine, Umeå University (contact via registrator@umu.se) for researchers who meet the criteria for access to confidential data.”

COMMENT: The authors say "No method was used to impute missing values". Why is that? Power is modest as it is (see next major comment) so why not try to at least paritally address that using multiple imputation? A clear explanation is needed.

RESPONSE: We have now added a section motivating why we did not use imputation of missing values (page 10, first paragraph): “In an elderly population, dropout from active study participation during long-term follow-up may be considerable due to health deterioration associated with aging. For the primary outcome events and fractures, follow-up was also performed among those who discontinued active participation, thereby avoiding the risk of attrition bias. Collection of the secondary outcome variables (OH and EQ5D) did, however, require active participation. We did not use imputation to replace values missing due to discontinuation, as it was reasonable to assume that discontinuation was often due to deterioration in health and that the available data were therefore not necessarily predictive of the missing data. A comparison of baseline characteristics and primary outcome events among those who discontinued active participation preterm (<3 years) to those who participated for at least 3 years is available in S2 Table.”

COMMENT: The results need to be interpreted more cautiously since the analyses are unerpowered to detect anything but very large effects. This is already discussed as limitation in the relevant section.

RESPONSE: The results are underpowered, and this has been discussed on page 19, first paragraph, page 20, first paragraph, page 23, first paragraph, and page 24, first paragraph.

COMMENT: Was the proportional hazards assumption evaluated in the Cox models?

RESPONSE: The proportional hazard assumption in the Cox models was not violated, as log(-log) plots did not indicate time-dependent effects (page 10, first paragraph): Log(−log) plots were used to evaluate the proportional hazard assumption in the Cox models.

COMMENT: More common to say "univariable" rather than "univariate" analyses.

RESPONSE: Univariate has been replaced with univariable (page 10, first paragraph): “Hazard ratios were calculated for the outcomes using univariable Cox proportional hazard regressions.”

COMMENT: The issue with the competing risks approach was not clear. I don't see why the competing risk of all-cause death could not be computed and if findings are similar, so be it (and for the best), as the authors imply.

RESPONSE: Besides using MACE as an outcome, an ambition was also to see if there might be a competition between cardiovascular death and other causes of death. It was, however, not possible to make a reliable analysis with solid results. Since cardiovascular death is a part of all-cause death, these two cannot compete.

COMMENT: The secondary prevention measures are limited by 2025 standards and dont include aspects such as prolonged dual anti-platelet therapy, low dose anti-coagulants, SGLTI and GLP1 analogues in diabetics and weight management. Were PCSK9I or inclisiran used in either group? Comments in discussion please.

RESPONSE: The patients were included and followed before these aspects of secondary prevention were introduced in the guidelines. This has now been addressed in the discussion (page 20, second paragraph): “The NAILED trial started recruitment in 2010, and the patients were followed until 2017. Therefore, the intervention and the control group participants likely did not receive the aspects of secondary cardiovascular prevention introduced in the guidelines after 2017. These include long-term secondary prevention with dual antiplatelet therapy, the lipid-lowering drugs proprotein convertase subtilisin/kexin type 9 inhibitors and inclisiran, and adding a glucagon-like peptide-1 receptor agonist or sodium-glucose cotransporter 2 inhibitor for patients with diabetes mellitus with established atherosclerotic cardiovascular disease (18).”

COMMENT: The effects of intervention will be dependent not only on the intervention efficacy, but on the standard treatment. Can the authors provide information on how these patients are managed in 'standard care'. Are targets for LDL and BP used in Sweden and how much intervention is typically provided by 'standard' primary care? In an 'excellent' primary care system, improvements may be hard to achieve.

RESPONSE: Usual care refers to secondary preventive treatment that the general practitioners usually have the primary responsibility for. A new section clarifying that all participants had the same target levels has been added (page 7, third paragraph): “The target BP and LDL-C levels were based on local guidelines, and the same target BP and LDL-C levels were used independently of group allocation. The target BP was <140/90 mmHg throughout the study period. The target LDL-C level was initially <2.5 mmol/L, but subsequent changes were made to match changes in local guidelines. On 31 March 2013, the target LDL-C level was lowered to <1.8 mmol/L for patients with diabetes, and this target level was adopted for all study participants on 1 January 2017 (21).”

A sentence regarding the performance of the control group in the NAILED trial was also added (page 19, first paragraph): “In the total NAILED trial, the intervention achieved a significant reduction in BP and LDL-C compared to usual care (19,20), despite the control group performing better than in the EUROASPIRE IV survey in the ACS cohort (20).”

COMMENT: To me, the effects of the intervention were very modest with a BP difference for sys and dia of only about 3-4 mmHg and an LDL difference of about 0.2 mmol/l. Can the authors provide data on how many patients achieved LDL targets of 1.8 (or 1.4) in each group? Would such small differences over a fairly short follow up period be expected to reduce events? For such small differences the study was probably under powered. Comment please in discussion.

RESPONSE: A section discussing the BP and LDL-C differences has been added (page 19, second paragraph): “During follow-up, the mean SBP, DBP, and LDL-C were approximately 5-6 mmHg, 3-4 mmHg, and 0.1-0.2 mmol/L, respectively, lower in the intervention group than in the control group. The intervention group also had a higher proportion of participants achieving target levels, and the follow-up time was longer than in similar trials (25,27). The BP reduction in the current trial can be compared to a meta-analysis of RCTs, where a reduction of SBP by 5 mmHg in patients with previous cardiovascular disease, with a mean age of 65 years, reduced the risk of major cardiovascular events by about 10% (2). The clinical relevance of the small numerical difference in LDL-C in the current trial is uncertain, but a reduction of LDL-C of 1 mmol/L has previously been shown to reduce the risk of major cardiovascular events in older adults by 26% (6).”

In Table 2, data on the proportion of patients who achieved target levels of BP and LDL-C during follow-up have been added.

COMMENT: Do we have any indirect measures of the efficacy of the lifestyle measures? Weight change in each group would be an important metric.

RESPONSE: Indirect measures of the efficacy of the lifestyle measures are not available because they were not part of the study's scope.

---

## [Decision Letter · Decision Letter 1]

19 Oct 2025

Nurse-led secondary preventive follow-up after stroke/TIA and ACS for patients aged 80 years or older: A post-hoc analysis of the randomized controlled NAILED trial

PONE-D-25-02444R1

Dear Dr. Ögren,

We’re pleased to inform you that your manuscript has been judged scientifically suitable for publication and will be formally accepted for publication once it meets all outstanding technical requirements.

Kind regards,

Giuseppe Andò, M.D., Ph.D.

Academic Editor

PLOS ONE

Additional Editor Comments (optional):

Reviewers' comments:

Reviewer's Responses to Questions

**Comments to the Author**

Reviewer #1: All comments have been addressed

Reviewer #2: All comments have been addressed

2. Is the manuscript technically sound, and do the data support the conclusions?

Reviewer #1: Yes

Reviewer #2: Yes

3. Has the statistical analysis been performed appropriately and rigorously?

Reviewer #1: Yes

Reviewer #2: Yes

4. Have the authors made all data underlying the findings in their manuscript fully available?

Reviewer #1: Yes

Reviewer #2: Yes

5. Is the manuscript presented in an intelligible fashion and written in standard English?

Reviewer #1: Yes

Reviewer #2: Yes

Reviewer #1: I am satisfied with the authors' responses and the resulting changes to the paper. I have nothing else to add.

Reviewer #2: All comments have been fully and clearly addressed. The authors have provided explanations to reviewer questions and have amended their MS appropriately. The paper now provides a full description of their stuidy population, methods and results. The discussion puts the results in context very well and illustrates the problems of prevention of CV events in this age group.

**Do you want your identity to be public for this peer review?** For information about this choice, including consent withdrawal, please see our Privacy Policy

Reviewer #1: No

Reviewer #2: **Yes: ** Jonathan N Townend

---

## [Editor Report · Acceptance letter]

PONE-D-25-02444R1

PLOS ONE

Dear Dr. Ögren,

I'm pleased to inform you that your manuscript has been deemed suitable for publication in PLOS ONE. Congratulations! Your manuscript is now being handed over to our production team.

Kind regards,

on behalf of

Prof. Giuseppe Andò

Academic Editor

PLOS ONE